# Fast Curable Polysiloxane-Silphenylene Hybrimer with High Transparency and Refractive Index for Optical Applications

**DOI:** 10.3390/polym13040515

**Published:** 2021-02-09

**Authors:** Kyungkuk Koh, Honglae Sohn

**Affiliations:** Department of Chemistry, Chosun University, Gwangju 61452, Korea; kkk890512@naver.com

**Keywords:** siloxane, fast curing, transmittance, refractive index, sol–gel

## Abstract

In this study, a fast curable polysiloxane-silphenylene hybrimer (PSH) was synthesized by the nonhydrolytic sol–gel condensation of phenyl-vinyl-oligosiloxane (PVO) and tris(dimethylhydrosilyl)benzene (TDMSB) under a Pt catalyst to investigate its optical property and thermal stability. The combination of PVO and tripod-type TDMSB results in a hybrimer with a fast curing time of 30 min. The PSH exhibited a high refractive index of 1.60, 1.59, and 1.58 at 450, 520, and 635 nm, respectively. High transmittance of 97% at 450 nm was obtained. The PSH exhibited a very high transmittance of 97% before thermal aging. The high optical transmittance of the PSH was only slightly decreased by 0.5% of the transmittance at 180 °C for 72 h after thermal aging, and high transparency was maintained almost constant even after 72 h of high-temperature treatment.

## 1. Introduction

Polymers with a high refractive index, excellent optical transparency, and high thermal stability have extensively been investigated in the field of advanced optical applications, such as organic light-emitting diodes (OLEDs), high-performance complementary metal–oxide–semiconductor (CMOS) image sensors, and microlens components of charge-coupled devices (CCD), due to their light extraction efficiency, long lifetime, and durability [1,2]. For the use of LEDs, a LED encapsulant as a protective material to prevent exposure to shock, heat, and moisture requires excellent optical characteristics with long-term thermal stability for commercial use. Since the light extraction efficiency of LEDs depends on the transparency and refractive index of the encapsulant, the molecular tailoring of conventional polymers is greatly desired.

The typical refractive index of conventional polymers is often in the range of 1.30–1.70. The mismatch of refractive indices between the semiconductors (*n*: 2.50–3.50) and the polymer encapsulants causes total internal reflection at certain angles and low light extraction efficiency in optoelectronic devices. Thus, it is more effective to have a high refractive index to improve light transmission efficiency. To increase the refractive indices of polymers, the introduction of substituents exhibiting low molar volumes and high molar refraction such as halogen or sulfur atoms and aromatic rings is one of the promising approaches [3]. Although commercial epoxy resins exhibit good visible light transmission and protection against gas, they are susceptible to yellowing at high temperatures and exhibit low refractive indices [4]. To overcome this issue, silicon-based resins and elastomers are investigated in the industry as encapsulants [5,6]. Polyphenylsiloxane [7] and epoxy-based polyphenylsiloxane materials have also been reported for their good transparency and deformation resistance at high temperatures [8]. Other various cross-linking agents, such as phenyl- and zirconium-based polysiloxane hybrid materials have shown good optophysical properties [9,10].

It is also important in the industry that the encapsulant has a high transmittance for a long duration as well as short curing times. Silicone resins containing silphenylene units are known materials for their high thermal degradation temperatures [11,12,13,14,15]. These characteristics are desirable for high-temperature coatings in semiconductor devices. Because of this property, silaryl-incorporated polysiloxanes can be applied to heat-resistant materials such as the LED encapsulants.

Herein, a polysiloxane-silphenylene hybrimer (PSH) was synthesized by using 1,3,5-tris(dimethylhydrosilyl)benzene (TDMSB) as a silarylene-containing dendrimer-type cross-linking agent. A siloxane-based phenyl-vinyl-oligosiloxane (PVO) was thermally cured with a cross-linker, TDMSB, via a hydrosilylation reaction over a Karstedt’s catalyst, platinum(0)-1,3-divinyl-1,1,3,3-tetramethyldisiloxane complex solution. This PSH showed an excellent transmittance property as well as ultrafast curing times.

## 2. Materials and Methods 

### 2.1. Materials

Dichlorodiphenylsilane (97%), 1,3,5-tribromobenzene, chlorodimethylsilane (97%), and hexaphenylcyclotrisiloxane (98%) were purchased from Alfa Aesar. Magnesium (powder, 99%) was purchased from Wako Pure Chemical Industries Ltd. Magnesium sulfate anhydride (MgSO_4_, 99%), sodium bicarbonate (NaHCO_3_, 99.5%), and organic solvents such as tetrahydrofuran (THF), toluene, diethyl ether, n-hexane, and xylene were purchased from OCI Co., Ltd. All organic solvents were distilled from sodium/benzophenone under argon gas. Barium hydroxide monohydrate (BH, Ba(OH)_2_·H_2_O, 98%) was purchased from Aldrich. The platinum(0)-1,3-divinyl-1,1,3,3-tetramethyldisiloxane complex solution (Karstedt’s catalyst) and vinyltrimethoxysilane (VTMS) were purchased from Sigma-Aldrich Co., Ltd. and used without further purification. 

### 2.2. Instrumentation

Transmittance was measured using a UV–Vis spectrometer (UV-2401 PC, Shimazu, Kyoto, Japan), and the refractive indices were recorded using a spectroscopic ellipsometer (M2000, Woollam, Lincoln, NE, USA), which measured the wavelength with the refractive index. The material hardness was measured using a durometer (GS-702N, Teclock, Nagano, Japan). The ^1^H and ^13^C NMR spectra were recorded using a Bruker AC-300 MHz spectrometer. FT-IR spectra were measured using a Nicolet model 5700 with a diffuse reflectance system (Spectra-Tech diffuse reflectance attachment, Thermo Scientific, Waltham, MA, USA). Thermal gravimetric analysis (TGA) was performed using a TGA-50H (Shimadzu, Kyoto, Japan) instrument under a nitrogen gas flow at a heating rate of 10 °C/min. Scanning electron microscopy (SEM; JSM-840A, Tokyo, Japan) was used to confirm the film thickness.

### 2.3. Synthesis of Diphenylsilanediol (DPSD)

First, sodium bicarbonate (6.52 g, 78.7 mmol) was dissolved in 800 mL of distilled water. The solution was then cooled to 0 °C, and dichlorodiphenylsilane (10 g, 39.5 mmol) was added dropwise. After the addition of dichlorodiphenylsilane was complete, the solution was stirred for 1 h. The resultant white product was extracted with diethyl ether, and the organic layer was washed successively with water and brine, followed by drying over magnesium sulfate. After filtration, the solvent was evaporated under reduced pressure, and the residue was recrystallized from toluene to afford 6.75 g (31.2 mmol, 80%) DPSD as white crystals: ^1^H NMR (300 MHz, CDCl_3_) δ 7.74–7.68 (m, 4H), 7.48–7.36 (m, 6H) 2.78 (m, 2H); ^13^C NMR (75 MHz, CDCl_3_) δ 134.5, 130.8, 128.2.

### 2.4. Synthesis of PVO

PVO was prepared according to a method described previously [9]. A 100 mL round-bottom flask equipped with a reflux condenser and a magnetic stirrer was charged with DPSD (10 g, 46.3 mmol), VTMS (10.28 g, 69.4 mmol), and BH (0.026 g, 0.1 mmol). The reaction mixture was refluxed at 80 °C for 24 h. After the reaction was complete, the reaction mixture was evaporated to remove methanol as a byproduct. In addition, BH was removed using a 0.30 μm pore-sized Teflon filter. The resultant product was obtained in a viscous oil form.

### 2.5. Synthesis of TDMSB

TDMSB was prepared according to a literature method [16]. Magnesium powder (4.7 g, 0.2 mol) was added to 20 mL of dried THF and stirred for 1 h under an argon atmosphere, followed by the dropwise addition of chlorodimethylsilane in dried THF (10 mL). After 30 min, 1,3,5-tribromobenzene in THF (30 mL) was added dropwise to the reaction mixture and stirred for 2 days. The solvent was removed under reduced pressure, and the salt was precipitated in n-hexane or pentane and filtered. The filtrate was subjected to silica gel chromatography with n-hexane and pentane. The product was obtained as a liquid (4.26 g, 48%). ^1^H-NMR (300 MHz, CDCl_3_) δ 7.73 (m, 3H), 4.46 (m, 3H), 0.36 (m, 18H); ^13^C-NMR (75 MHz, CDCl) δ 144.5, 143.5, 140.7, 139.9, 139.8, 131.2.

### 2.6. Synthesis of the PSH

The PSH was synthesized by the hydrosilylation reaction of PVO and TDMSB in the presence of Karstedt’s catalyst. The PVO (2.00 g, 5.3 mmole of Si(CH=CH_2_)(OSiPh_2_)_3/2_ unit) was mixed with TDMSB (0.45 g, 1.8 mmol), and a 0.1 wt % of Pt Karstedt’s catalyst in xylene (~2% of Pt) was added. The volatile was subsequently removed under reduced pressure, and the mixture was cast in a glass mold composed of a surface-treated with hexaphenylcyclotrisiloxane (HPTS), which allowed for easy detachment of the PSH sample from the glass mold. The samples were heat-cured at 180 °C under ambient atmosphere for 30 min. 

## 3. Results and Discussion

In this research, a silphenylene-containing polyphenylsiloxane hybrimer (silphenylene hybrimer) was synthesized with polyphenylsiloxane (PVO) and trihydrosilylbenzene (TDMSB) as the cross-linker. This silphenylene hybrimer (PSH) exhibits a high refractive index of 1.6 and a high transmittance of 97% at 450 nm. Scheme 1A shows the synthesis of PVO via sol–gel condensation of methoxysilane and silanol. Methoxy (–OCH_3_) groups of VTMS reacted with silanol (Si–OH) groups of DPSD via nucleophilic attack by deprotonation of the hydroxyl (–OH) groups, forming covalently bridged siloxane (Si–O–Si) bonds [17,18,19,20,21]. Because of the phenyl group cleavage under acidic conditions, barium hydroxide (basic catalyst) was used to promote condensation. Under a nonhydrolytic sol–gel process, the silane monomers were directly condensed to form oligomers without water, which must be avoided to improve resin stability [21,22]. PVO was obtained as a colorless oil and confirmed by ^1^H and ^13^C NMR. 

The PSH shown in Scheme 1B was synthesized via the hydrosilylation reaction of the sol–gel-synthesized PVO resin with TDMSB as the cross-linker over a Pt catalyst. Herein, the unit molar ratio between the vinyl groups of PVO and the hydrosilyl groups of TDMSB was optimized to be 1:1. The content of the catalyst was optimized to be 0.1 wt % of the mixed PVO and TDMSB. The hydrosilylation reaction was performed at 180 °C in air for 30 min. FT-IR spectra of the as-prepared PSH show the curing behavior of the PVO and TDMSB with the Pt catalyst (Figure 1). The band at 2120 cm^−1^ was assigned to Si-H groups in the TDMSB and those at 1600 cm^−1^ were assigned to the vinyl group in the PVO resin. After thermal curing, these bands completely disappeared and were no longer present after the reaction due to cross-linking between the vinyl and hydrosilane groups in the resin. Based on these analyses, the hydrosilylation of PVO and TDMSB was fully completed within 0.5 h.

Since heterogeneity is known to be an inherent property of polymer networks, one of the ultimate goals in polymer science is homogeneity in an ideal polymer network. To obtain good homogeneity, rate control over the reaction conversion and fluidity of the pregel solution are important to allow for the homogeneous mixing of prepolymers and cross-linking agents and to prevent the formation of an underdeveloped network with defects such as clusters and dangling chains. In order to investigate the effect of homogeneity and cross-linking density properties, we measured the FT-IR and transmittances at several stoichiometric ratios, which are the unit molar ratio between the vinyl groups of PVO and the hydrosilyl groups of silphenyl. The maximum value of the transmittance in the optical spectrum and minimum vibrational intensities of the Si-H and vinyl groups in the FT-IR spectrum were obtained as equimolar reactants reacted with each other. The optical transmittance of the PSH was near 100% at equimolar reactants, indicating that the PSH has an optimum homogeneous network structure and high cross-linking density. It is interesting to note that the optical transmittance of the polyphony hybrimer (PPH) was about 90% when the hydrosilyl groups of the cross-linking agent were nonsymmetrical and localized [9]. This steric hindrance resulted in an optimum molar ratio between phenyltri(dimethylsiloxy)silane and PVO to be 1.25:1, leading to a less homogenous and incomplete network structure in the PPH [9]. 

Since the DSC data for the phenyl hybrimer indicated that the reaction showed an exothermic cross-linking reaction by hydrosilylation [9], the fabrication of the PSH would be expected to also be an exothermic reaction. The thermomechanical properties of the PPH, which are similar to those of the PSH, are reported, showing that the PPH exhibits a high storage modulus and glass transition temperature compared to the commercial polydimethylsiloxane (PDMS) resin. The thermomechanical properties of the PSH are quite soft and bendable, which is desirable for LED encapsulation applications for the protection between the PCB and LED, which can be easily damaged due to mechanical stress at high temperatures.

Figure 2A shows the hardness (Shore D) of the PSH according to the curing times. The Shore D of 70 of the PSH was reached within 30 min and then the hardness of the PSH was slowly increased up to 77.2. For the long-term reliability of the LED encapsulant, it should exhibit high thermal stability to prevent heat-related decomposition (as measured by TGA) as well as resistance to yellowing induced by heat. TGA analysis of the silphenyl hybrimer was measured for the long-term reliability of the encapsulant, shown in Figure 2B. The temperature of 5% weight loss for the PSH was 350 °C. This high thermal stability and short curing time result of the PSH originated from high cross-linking between the multifunctional silphenyl molecule and polyvinylsiloxane. 

LED efficiency can be enhanced by using high-quality encapsulants and improving thermal stability as well as increasing the refractive index. Polysiloxane-based hybrimers have been recently developed to have a high refractive index to improve light transmission efficiency. Figure 3 shows the previously reported cross-linking units for the polysiloxane hybrimers, such as a polyphenyl hybrimer (PPH), an epoxy hybrimer (PES), and a zirconium hybrimer (PZH) [8,9,10]. Since the hydrosilyl groups of the cross-linking agent of phenyltri(dimethylsiloxy)silane in phenyl hybrimer are localized and directed to one side, the hydrosilylation reactions take place inefficiently. However, the silphenylene of the silphenyl hibrimer reported in this study is a dendrimer-type cross-linking agent, o the hydrosilyl groups of which are delocalized around the benzene moiety and exhibit a very fast curing process of less than 0.5 h. This fast curing process originates from high-density cross-linking between PVO and silphenylene. Figure 4 compares the hardness values (Shore D) of polysiloxane resin (PS), PPH, PES, PZH, PSH (this work), and OE-6630 (a commercial encapsulant from Dow Corning Corporation) along with their curing times. As expected, the PSH exhibits the fastest curing rate due to the delocalized multifunctionality over the benzene moiety. The hardness of the PSH was 77 Shore D, which indicated that the high-density cross-linking process took place efficiently for the generation of the hybrimer. The silphenyl hybrimer is harder than the phenyl hybrimer and commercial silicon encapsulant such as OE-6630 (Dow Corning, Shore D = 40) [8]. The increase in hardness is due to the rigid structure of the phenyl group of the PSH. 

The mismatch of the refractive index values between the semiconductor (*n* = 2.50–3.50) and the polymer encapsulants (*n* = 1.40–1.60) often causes low light extraction efficiency of the device in high-brightness LED fabrications due to total internal reflection when light travels from the semiconductor into the encapsulant at certain incident angles. A recent issue for LED encapsulation materials requires a high refractive index with antiyellowing at a high junction temperature. Epoxy- and siloxane-based polymers, which are common commercial products, are generally used for LED encapsulation, but their refractive index is too low in encapsulant applications. To increase the refractive index of polymers, various types of phenyl-containing polysiloxane resin have recently been developed since the phenyl groups in polymers are highly polarizable. Its refractive index is about ~1.52 [9]. Until now, phenyl-based polysiloxane materials with a high refractive index (over 1.6) and high transparency have not been reported. 

The optical property of silphenyl hybrimer was investigated for use as an encapsulant. The transmittance and refractive index of the cured silphenyl hybrimer were measured as they are important for determining encapsulant quality. The refractive index of the prepared phenyl hybrimer was assessed using a spectroscopic ellipsometer, as shown in Figure 5. The PSH exhibited a high refractive index of 1.60, 1.59, and 1.58 at 450, 520, and 635 nm. The transmittance of the silphenyl hybrimer was examined using a UV–Vis-NIR spectrometer (Figure 6). The silphenyl hybrimer exhibited a very high transmittance (97% at 450 nm), comparable to other encapsulants synthesized by hydrosilylation.

Figure 6 compares the refractive index and transmittance for the PS, PPH, PES, PZH, PSH, and OE-6630 (Dow Corning). The refractive index and transmittance of the PSH were 1.60 and 97% at 450 nm, respectively. The PSH showed the highest refractive index and transmittance among the other hybrimers. Since the temperature at the junction in the LED device can be up to 120 °C, the thermal stability of the polymer materials for LED encapsulation over 150 °C is required for device reliability and long-term lifetime. Phenyl-based polymer materials are unstable against yellowing at high temperatures because the phenyl groups and uncrosslinked vinyl groups in the polymers can be a source of oxidation and discoloration by heat for long periods. However, inorganic siloxane organic phenyl hybrid materials in the PSH exhibited very high transparency accompanied by resistance to yellowing induced by thermal degradation above 180 °C aging in air. To confirm the good thermal stability of the prepared materials in terms of optical degradation, the transmittance before and after thermal aging at 180 °C for 72 h was measured (Figure 7). The thermal aging test for the PSH was measured at 180 °C for 72 h to compare its resistance to thermal degradation. In Figure 7A, the transmittance spectra of the PSH were obtained before and after thermal aging at a normal wavelength range of 350–700 nm at every 24 h. The PSH exhibited a very high transmittance of 97% before thermal aging. The high optical transmittance of the PSH was only slightly decreased by 0.5% of the transmittance at 180 °C for 72 h after thermal aging (Figure 7B), and high transparency was maintained almost constant even after 72 h of high-temperature treatment. The highest thermally stable silicone resin that is commercially available for a high refractive index is Shin-Etsu (KER6100/CAT-PH), which provides a yellowing temperature up to 120 °C for 1000 h and a refractive index of ~1.52. Therefore, the results of the thermal stability tests for the PSH suggested that the novel silphenyl hybrimer has great potential as a high-performance LED encapsulant with long-term reliability. 

## 4. Conclusions

In summary, the fabrication and optical characterization of a novel silphenyl hybrimer synthesized from PVO and TDMSB was reported. The resulting polysiloxane silphenyl hybrimer exhibited a high refractive index of 1.58 and an excellent transmittance of 97% at 450 nm. The silphenylene of silphenyl hibrimer exhibited a very fast curing process within 0.5 h due to the even distribution of the hydrosilyl groups in the benzene moiety. Long-duration exposure (up to 72 h) to high temperature was tested to examine the feasibility of real-life applications as part of LED assemblies. After thermal curing at 180 °C for 72 h, the transmittance and refractive index remained almost constant, which are the most important optical properties for the photoextraction efficiency of LED encapsulants.

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
