# Peer review of "Fast Curable Polysiloxane-Silphenylene Hybrimer with High Transparency and Refractive Index for Optical Applications"

_polymers, 2021, doi:10.3390/polym13040515_

Round 1
Reviewer 1 Report
The manuscript deals with the preparation and characterization of a new type of polymers with high refractive index, excellent optical transparency, high thermal stability, excellent transmittance property and ultrafast curing times. These polymer-based materials are very interesting for several kinds of optical devices and applications.
The paper shows a basic research work, with standard methods for the characterization of the materials. The synthesis of the materials is not new. In this respect the paper has a middle-low value and it is more a routine work.
Additional experiments and, more important, additional results discussion are necessary to improve the quality and scientific level of the research work.
For instance, authors should try to anser or to complete the following items:
- The characterization of the polymer materials is based on FT-IR and NMR, what about DSC, thermogravimetric and thermomechanic properties? Do the authors have an idea about these properties?
- Are there any information about the homogeneity of the polymer structure?
- Did the authors try to change the relative percentage contributions of different components, in particular, the crosslinker percentage?
- The particular values of refractive index measured in the presented work can be explained in terms of the particular polymer properties (such as density, density of crosslinkers, molecular structure, ...)?
- Concerning the thermal stability and aging effects, what are the mechanisms explaining the presented results.
Author Response
The characterization of the polymer materials is based on FT-IR and NMR, what about DSC, thermogravimetric(TGA) and thermomechanic properties? Do the authors have an idea about these properties?
Ans.) The following sentences are added and revised in the manuscripts for the answer.
“Since the DSC data for the phenyl hybrimer indicated that the reaction showed an exothermic cross-linking reaction by hydrosilylation, the fabrication of the PSH would be expected to be also exothermic reaction. The thermomechanic properties of PPH, which is similar to that of PSH, are reported that the PPH exhibits a high storage modulus and glass transition temperature compared to the commercial polydimethylsiloxane (PDMS) resin. The thermomechanic properties of PSH are quite soft and bendable, which is desirable for LED encapsulation applications for the protection between the PCB and LED, which can be easily damaged due to mechanical stress at high temperatures. The highest thermally stable silicone resin which is commercially available for high refractive index is Shin-Etsu (KER6100/CAT-PH), which provides a yellowing temperature up to 120 ℃ for 1000 h and a refractive index of ∼1.52.”
- Are there any information about the homogeneity of the polymer structure?
- Did the authors try to change the relative percentage contributions of different components, in particular, the crosslinker percentage?
Ans.) For those two questions, the following sentences are added and revised in the manuscripts.
“Since heterogeneity is known to be an inherent property of polymer networks, one of the ultimate goals in polymer science is the homogeneity in an ideal polymer network. To obtain the good homogeneity, the rate control over reaction conversion and the fluidity of the pregel solution are important to allow for the homogeneous mixing of prepolymers and cross-linking agents and to prevent the formation of an underdeveloped network with defect such as clusters and dangling chains. In order to investigate the effect of homogeneity and crosslinking density property, we measured the FT-IR and transmittances at several stoichiometric ratios, which are the molar ratios of PVO to silphenyl. The maximum value of the transmittance in optical spectrum and minimum vibrational intensities of Si-H and vinyl group in FT-IR spectrum were obtained at equimolar reactants reacted with each other. The optical transmittance of PSH was almost 100% at equimolar reactants, indicating that the PSH has an optimum homogeneous network structure and high crosslinking density. It is interesting to note that the optical transmittance of PPH was about 90%, when the hydrosilyl groups of cross-linking agent were unsymmetrical and localized [9]. This steric hindrance results in the optimum molar ratio between phenyltri(dimethylsiloxy)silane and PVO to be 1.25:1, leading less homogenous and incomplete network structure and in PPH [9].
- The particular values of refractive index measured in the presented work can be explained in terms of the particular polymer properties (such as density, density of crosslinkers, molecular structure, ...)?
Ans.) For this question, the following sentences are added and revised in the manuscripts.
“The mismatch of the refractive index values between the semiconductor (n = 2.50 ~ 3.50) and the polymer encapsulants (n = 1.40 ~1.60) often causes low light extraction efficiency of the device in high-brightness LED fabrications due to total internal reflection, when light travels from the semiconductor into the encapsulant at certain incident angles. A recent issue for the LED encapsulation materials requires a high refractive index with anti-yellowing at high junction temperature. Epoxy- and siloxane-based polymers, which are the common commercial products, are generally used for LED encapsulation, but their refractive index is too low in encapsulant applications. To increase refractive index of polymers, various types of phenyl involved polysiloxane resin are recently being developed, since the phenyl groups in polymers are highly polarizable. Its refractive index is about ~1.52 [9]. Until now, the phenyl-based polysiloxane materials with high refractive index (over 1.6) and high transparency have not been reported.”
- Concerning the thermal stability and aging effects, what are the mechanisms explaining the presented results.
Ans.) The following sentences are added and revised in the manuscripts.
“Since the temperature at junction in LED device is up to 120°C, the thermal stability of the polymer materials for LED encapsulation over 150°C is required for device reliability and long term lifetime. The phenyl-based polymer materials are unstable against yellowing at high temperatures because the phenyl groups and un-crosslinked vinyl groups in the polymers can be the source of oxidation and discoloration by heat for long periods. However, inorganic siloxane–organic phenyl hybrid materials in PSH exhibited very high transparency accompanied by resistance to yellowing induced by thermal degradation above 180°C aging in air.”

Reviewer 2 Report
This work presents the synthesis of polysiloxane-silphenylene hybrimer (PSH) from phenyl-vinyl-oligosiloxane (PVO) and tris(dimethylhydrosilyl)benzene (TDMSB). 1H NMR, 13C NMR and FT-IR spectroscopy was used to confirm the chemical structure of the synthesized compounds. The investigation of optical properties (transmittance and refractive index), hardness, and thermal stability of the obtained cross-linked polymer samples was performed.
The following questions and comments can be addressed to the Authors of this manuscript:
- Why the temperature of180°C was selected for the synthesis and thermal aging of PSH, as the TGA curve (Fig. 2B) shows that the thermal decomposition of PSH starts at 150°C.
- Is the thermal stability up to 150°C sufficient for the proposed application of PSH in LED encapsulation?
- The high cross-linking between silphenyl molecule and polyvinylsiloxane is declared in the Results and Discussion part (lines 146,147), however the cross-linking density was not determined. High thermal stability of compounds can be caused not only by the cross-linking density but also by the highly aromatic structure of compounds. Moreover, the bulky substituents of PVO and close position of TDMSB functional groups suggest that a lot of functional groups had to remain unreacted due to the steric hindrances.
- Comparison of TGA curves, gel fraction and swelling of PSH samples before and after thermal aging at 180°C would provide more information about cross-linking density and influence of selected temperature (180°C) on the structure of the obtained polymers.
Author Response
The following questions and comments can be addressed to the Authors of this manuscript:
- Why the temperature of180°C was selected for the synthesis and thermal aging of PSH, as the TGA curve (Fig. 2B) shows that the thermal decomposition of PSH starts at 150°C.
Ans.) The following sentence is added and revised in the manuscripts.
“Since the silphenyl hybrimers began to degrade at 180°C, a very high temperature was selected for the fabrication and thermal aging of silphenyl hybrimers.”
- Is the thermal stability up to 150°C sufficient for the proposed application of PSH in LED encapsulation?
Ans.) The following sentence is added and revised in the manuscripts.
“Since the temperature at junction in LED device is up to 120°C, the thermal stability of the polymer materials for LED encapsulation over 150°C is required for device reliability and long term lifetime.”
- The high cross-linking between silphenyl molecule and polyvinylsiloxane is declared in the Results and Discussion part (lines 146,147), however the cross-linking density was not determined. High thermal stability of compounds can be caused not only by the cross-linking density but also by the highly aromatic structure of compounds. Moreover, the bulky substituents of PVO and close position of TDMSB functional groups suggest that a lot of functional groups had to remain unreacted due to the steric hindrances.
Ans.) The following sentence is added and revised in the manuscripts.
“In order to investigate the effect of homogeneity and crosslinking density property, we measured the FT-IR and transmittances at several stoichiometric ratios, which are the molar ratios of PVO to silphenyl. The maximum value of the transmittance in optical spectrum and minimum vibrational intensities of Si-H and vinyl group in FT-IR spectrum were obtained at equimolar reactants reacted with each other. The optical transmittance of PSH was near 100% at equimolar reactants, indicating that the PSH has an optimum homogeneous network structure and high crosslinking density. It is interesting to note that the optical transmittance of PPH was about 90%, when the hydrosilyl groups of cross-linking agent were unsymmetrical and localized [9]. This steric hindrance results in the optimum molar ratio between phenyltri(dimethylsiloxy)silane and PVO to be 1.25:1, leading less homogenous and incomplete network structure and in PPH [9].”
- Comparison of TGA curves, gel fraction and swelling of PSH samples before and after thermal aging at 180°C would provide more information about cross-linking density and influence of selected temperature (180°C) on the structure of the obtained polymers.
Ans.) Comparison of TGA before and after thermal aging will provide good information about cross-linking density. However, to investigate the crosslinking density property, we synthesized PSH samples at several stoichiometric ratios and found the optimum conditions. The PSH samples show almost no decrease of transmittance after thermal aging, which indicates the high cross-linking density. This could be an alternative way to prove it.

Round 2
Reviewer 1 Report
The authors have answered to the main points and the manuscript has been modified taking into account the reviewers' comments. However, the paper can still be improved by adding 1) a state of the art of the polymer chemistry materials and synthetic strategies used and reported in the literature for similar systems; 2) a state of the art of polymers with high refractive index. This is important to enlarge the readers audience and to include a more appropriate and complete bibliography.
Reviewer 2 Report
The manuscript was corrected according to the comments.